EMBO
Molecular Medicine

# Inhibition of Aurora Kinase B attenuates fibroblast activation and pulmonary fibrosis

Rajesh K Kasam[1,2], Sudhir Ghandikota[3,4] (ID), Divyalakshmi Soundararajan[1], Geereddy B Reddy[2], Steven K Huang[5], Anil G Jegga[3,4,6,*] (ID) & Satish K Madala[1,6,**] (ID)

## Abstract

Fibroblast activation including proliferation, survival, and ECM production is central to initiation and maintenance of fibrotic lesions in idiopathic pulmonary fibrosis (IPF). However, druggable molecules that target fibroblast activation remain limited. In this study, we show that multiple pro-fibrotic growth factors, including TGFα, CTGF, and IGF1, increase aurora kinase B (AURKB) expression and activity in fibroblasts. Mechanistically, we demonstrate that Wilms tumor 1 (WT1) is a key transcription factor that mediates TGFα-driven AURKB upregulation in fibroblasts. Importantly, we found that inhibition of AURKB expression or activity is sufficient to attenuate fibroblast activation. We show that fibrosis induced by TGFα is highly dependent on AURKB expression and treating TGFα mice with barasertib, an AURKB inhibitor, reverses fibroblast activation, and pulmonary fibrosis. Barasertib similarly attenuated fibrosis in the bleomycin model of pulmonary fibrosis. Together, our preclinical studies provide important proof-of-concept that demonstrate barasertib as a possible intervention therapy for IPF.

**Keywords** Aurora Kinase B; Barasertib; fibroproliferation; pulmonary fibrosis; Wilms' tumor 1

**Subject Categories** Pharmacology & Drug Discovery; Respiratory System

## Introduction

Idiopathic pulmonary fibrosis (IPF) is a chronic and progressive fibrotic lung disease characterized by increased proliferation and survival of mesenchymal cells that are involved in excessive production and deposition of extracellular matrix (ECM) proteins in the distal areas of the lung (Wynn & Ramalingam, 2012; Barratt *et al*, 2018). The median survival rate of IPF patients is 3–5 years from diagnosis, and IPF claims more lives annually than many types of cancer (Gribbin *et al*, 2006; Ley *et al*, 2011). The increasing burden of IPF is not simply reflective of an aging population, as age-adjusted mortality for IPF is increasing as well (Pardo & Selman, 2016). Currently, two FDA-approved drugs, Ofev (nintedanib) and Esbriet (pirfenidone), are available for the treatment of patients with IPF. However, neither of these drugs provide a cure, and both of them are associated with several serious drug-related side effects (King *et al*, 2014; Richeldi *et al*, 2014; Hughes *et al*, 2016). Pirfenidone and nintedanib both cause gastrointestinal events, while pirfenidone also causes skin disorders including rash and photosensitivity (Costabel *et al*, 2014; Wollin *et al*, 2015; Lancaster *et al*, 2017; Kasam *et al*, 2019). Intolerance to these medicines and their variable efficacy justify the need to identify other therapeutic alternatives.

In the pathogenesis of IPF, lung-resident fibroblasts have been shown to proliferate and differentiate into contractile mesenchymal cells, called myofibroblasts, which participate in continuous ECM production, secretion, and deposition in distal areas of the lung (Phan, 2008; Sontake *et al*, 2017). Multiple cytokines and growth factors released by immune cells, fibrocytes, and the lung epithelium are known to enhance the fibrogenic activity of lung-resident fibroblasts (Baughman *et al*, 1999; Allen & Spiteri, 2002). Published reports indicate that pathological processes in IPF are often stimulated by 1) receptor tyrosine kinases activated by ligands of the epidermal growth factor receptor (EGFR), as well as platelet-derived growth factor (PDGF), connective tissue growth factor (CTGF), insulin-like growth factor 1 (IGF1), basic fibroblast growth factor (FGF), and vascular endothelial growth factor (VEGF) receptors; 2) SMAD and non-SMAD (PI3K and MAPK) signaling pathways activated by TGFβ and PDGF receptors; and 3) STAT3 and STAT6 signaling pathways activated by IL-6 cytokines (IL-6 and IL-10) and Th2 cytokines (IL-4 and IL-13), respectively (Wilson & Wynn, 2009; Wynn, 2011; Madala *et al*, 2012; Prele *et al*, 2012; Grimminger *et al*, 2015; Walton *et al*, 2017). These data illustrate several unanswered

1  Division of Pulmonary Medicine, Cincinnati Children's Hospital Medical Center, Cincinnati, OH, USA
2  Department of Biochemistry, National Institute of Nutrition, Hyderabad, India
3  Division of Biomedical Informatics, Cincinnati Children's Hospital Medical Center, Cincinnati, OH, USA
4  Department of Computer Science, University of Cincinnati College of Engineering, Cincinnati, OH, USA
5  Division of Pulmonary and Critical Care Medicine, Department of Internal Medicine, University of Michigan Medical School, Ann Arbor, MI, USA
6  Department of Pediatrics, College of Medicine, University of Cincinnati, Cincinnati, OH, USA
   *Corresponding author. Tel: +1 513 636 0261; E-mail: anil.jegga@cchmc.org
   **Corresponding author. Tel: +1 513 636 9852; E-mail: satish.madala@cchmc.org

questions, but also suggest a possible redundant function and synergism between multiple cytokines and fibrogenic pathways. In support of this, inhibition of a single pathway or growth factor in mouse models or human clinical trials has not been effective in reversing established and ongoing fibrosis in the lung (Chaudhary *et al*, 2007; Vallath *et al*, 2014; Raghu, 2017). As fibrosis is likely heterogeneous in etiology and molecular pathophysiology, attempts to block or counteract single upstream or downstream pathways may not be sufficient to inhibit cellular processes associated with ongoing fibrosis. Therefore, identification of converging or overlapping pathways downstream of growth factors involved in fibroblast activation could result in more effective therapies. In a recent study, we used an unbiased chemical genomics approach to identify drug targets that regulate multiple pro-fibrotic processes including fibroproliferation, migration, invasiveness, and ECM production (Sontake *et al*, 2017). This novel *in silico* screening platform has identified several small molecule inhibitors including inhibitors of Hsp90, tyrosine kinases, and mitotic kinases, whose gene expression profiles were inversely correlated with differentially expressed gene networks in IPF lungs.

Aurora kinase B (AURKB) is a ubiquitously expressed mitotic kinase that belongs to the family of serine/threonine kinase (STKs) (Nigg, 2001; Fu *et al*, 2007). AURKB regulates multiple key events during cell division, which includes, but is not limited to chromatin condensation, chromosome bi-orientation, and cytokinesis (Kollareddy *et al*, 2008; Lampson & Cheeseman, 2011). Moreover, recent studies have identified interphase functions of AURKB during various pathophysiological settings (Wu *et al*, 2011; Gully *et al*, 2012; Frangini *et al*, 2013). It is well-documented that overexpression of AURKB in numerous cancer types causes uncontrollable tumor cell proliferation and helps in evading apoptotic death (Sorrentino *et al*, 2005; Jha *et al*, 2013; Tang *et al*, 2017). The role of AURKB in fibroblast activation in the pathogenesis of IPF has remained unexplored. We had previously shown that WT1 is a critical transcription factor that promotes fibrogenesis (Sontake *et al*, 2015, 2018). We recently performed RNA-seq in fibroblasts for which WT1 was knockdown, and AURKB was identified as a major target of WT1 (Sontake *et al*, 2018). Here, we show for the first time that AURKB is upregulated in fibroblasts of IPF and a mouse models of pulmonary fibrosis. Multiple growth factors such as TGFα, CTGF, and IGF1 upregulate AURKB in lung fibroblasts. WT1 acts as a key transcription factor involved in TGFα-induced AURKB expression in fibroblasts. In support of its pro-fibrotic functions, we found that inhibition of AURKB expression or activity attenuated proliferation, survival, and ECM production in lung fibroblasts. *In vivo* studies using the TGFα- and bleomycin-induced mouse models of pulmonary fibrosis demonstrate that barasertib, a known AURKB inhibitor, attenuates fibroproliferation, myofibroblast survival, and ECM deposition in established and ongoing pulmonary fibrosis.

# Results

## AURKB is upregulated in IPF fibroblasts

To determine the fibroproliferative effects of multiple pro-fibrotic growth factors on AURKB expression, we treated primary fibroblasts isolated from human lungs with multiple growth factors including TGFα, TGFβ, CTGF, and IGF1. We found a significant increase in the expression of AURKB in fibroblasts treated with TGFα, CTGF, and IGF1 but none with TGFβ (Fig 1A). To determine whether AURKB is overexpressed in IPF fibroblasts, we isolated fibroblasts from the lungs of patients with IPF and non-fibrotic controls and measured the amount of AURKB transcripts. We observed a significant increase in AURKB transcripts in fibroblasts of IPF compared to non-IPF controls (Fig 1B). To determine the expression of AURKB in the lungs of IPF, we immunostained lung sections with antibody against AURKB and observed a marked increase in AURKB staining in spindle-shaped fibroblasts located in subpleural regions and fibrotic foci of IPF lung tissue compared to normal lungs (Fig 1C).

## AURKB is upregulated during TGFα- and bleomycin-induced pulmonary fibrosis

To determine whether AURKB is upregulated during TGFα-induced pulmonary fibrosis, Western blot analysis was performed on total lung lysates from control (CCSP/-) and TGFα (CCSP/TGFα) mice on Dox for 6 weeks using antibodies against AURKA and AURKB. We observed a significant increase in the protein levels of AURKB but not AURKA in TGFα mice compared to controls (Fig 1D). In support, we observed increased staining for AURKB in the nucleus of spindle-shaped cells localized in the subpleural fibrotic lesions of TGFα mice on Dox for 6 weeks, but limited or no staining in the lungs of control mice (Fig 1E). Further, we found that the majority of AURKB is localized in the nucleus of vimentin-positive fibroblasts (Fig EV1). Similarly, we observed a significant increase in AURKB protein levels in the lungs of bleomycin-treated mice compared to saline-treated control mice (Fig 1F).

## Wilm's tumor 1 regulates the AURKB expression

Recent studies from our laboratory have shown that WT1 is upregulated in mesenchymal cells of subpleural fibrotic lung lesions and that it contributes to pulmonary fibrosis pathogenesis (Sontake *et al*, 2015). To test whether WT1 affects AURKB expression, we knocked down WT1 expression using WT1-specific siRNA in fibroblasts isolated from IPF lungs or TGFα mice on Dox for 4 weeks. Compared to control siRNA, knockdown of WT1 was sufficient to attenuate the expression of AURKB in lung fibroblasts isolated from both IPF and a mouse model of TGFα-induced pulmonary fibrosis (Fig 2A and B). We next assessed the effect of adenovirus-mediated overexpression of WT1 on AURKB levels in fibroblasts. Overexpressing WT1 in fibroblasts led to a significant increase in both transcripts and protein of AURKB (Figs EV2A and 2C). Knockdown of WT1 in fibroblasts also abolished the ability of TGFα to increase AURKB expression (Fig 2D). To determine whether WT1 is a transcription factor that directly binds to *AURKB*, we performed a computational analysis of the promoter region of human and mouse *AURKB* and found the presence of conserved WT1 binding sites (Fig EV2B). We performed chromatin immunoprecipitation (ChIP) analysis using primary lung-resident fibroblasts isolated from TGFα mice on Dox for 8 weeks. ChIP analysis using WT1-specific antibody compared with control isotype antibody revealed that WT1 bound to the *AURKB* gene, suggesting the role of WT1 in AURKB

expression (Fig 2E). We next overexpressed WT1 in HEK293 cells co-transfected with AURKB promoter fused with luciferase reporter and observed increased luciferase activity compared to cells transfected with control plasmids (Fig 2F). These results support that WT1 functions as a positive regulator of AURKB by directly binding to the AURKB promoter in fibroblasts.

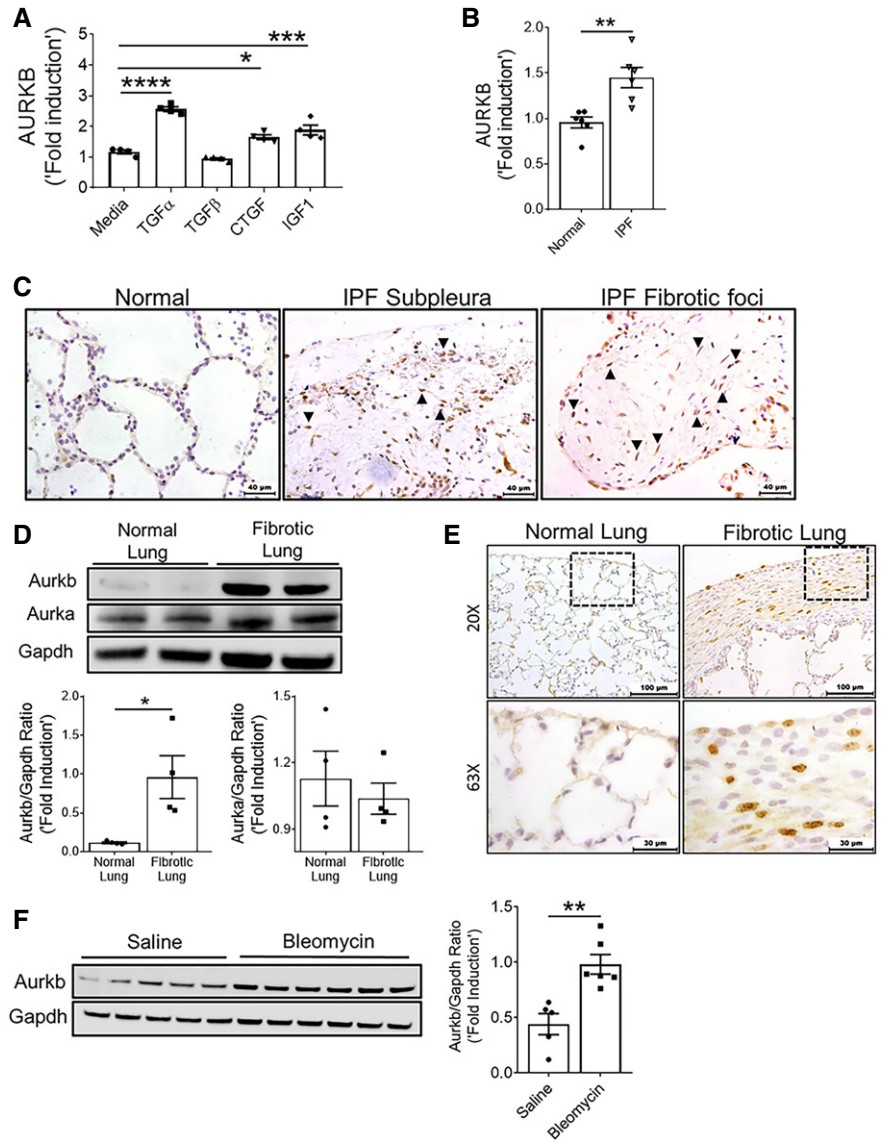

**Figure 1. Multiple growth factors upregulate AURKB in severe fibrotic lung disease.**

A   Quantification of AURKB transcripts in non-IPF lung-derived resident fibroblasts treated with indicated mitogens for 16 h. *$P < 0.05$, ***$P < 0.0005$, and ****$P < 0.00005$, 1-way ANOVA ($n = 4$).

B   Quantification of AURKB gene transcripts in isolated fibroblasts from non-IPF and IPF lung stromal cell cultures. **$P < 0.005$, unpaired $t$-test ($n = 6$).

C   IPF and non-IPF lung sections were immunostained using AURKB antibody ($n = 4$). Representative images were obtained at 40× magnification. Scale bar: 40 μm. Dotted area: black; myofibroblastic core, blue; active fibrotic front. Arrows indicate AURKB positive cells.

D   Immunoblots of Aurkb and Aurka in total lung lysates of normal (CCSP/−) and fibrotic (CCSP/TGFα) mice fed with Dox for 6 weeks. Quantification was performed using phosphor imager software and normalization was done using loading control Gapdh. *$P < 0.05$, unpaired $t$-test ($n = 4$).

E   Immunostaining was performed using AURKB antibody in lung sections of control (CCSP/−) and TGFα (CCSP/TGFα) mice on Dox for 6 weeks ($n = 6$). Representative images were obtained at 20× (low; Scale bar: 100 μm) and 63× (high; Scale bar: 30 μm) magnification. Dashed box indicates area of the section showing in high magnification.

F   Western blot analysis of Aurkb in total lung lysates from saline and bleomycin-treated mice. Quantification was performed using phosphor imager software, and normalization was done using loading control Gapdh. **$P < 0.005$, unpaired $t$-test ($n = 4$–5).

Data information: All data were presented as mean ± SEM. Exact $P$ values are shown in Appendix Table S6.
Source data are available online for this figure.

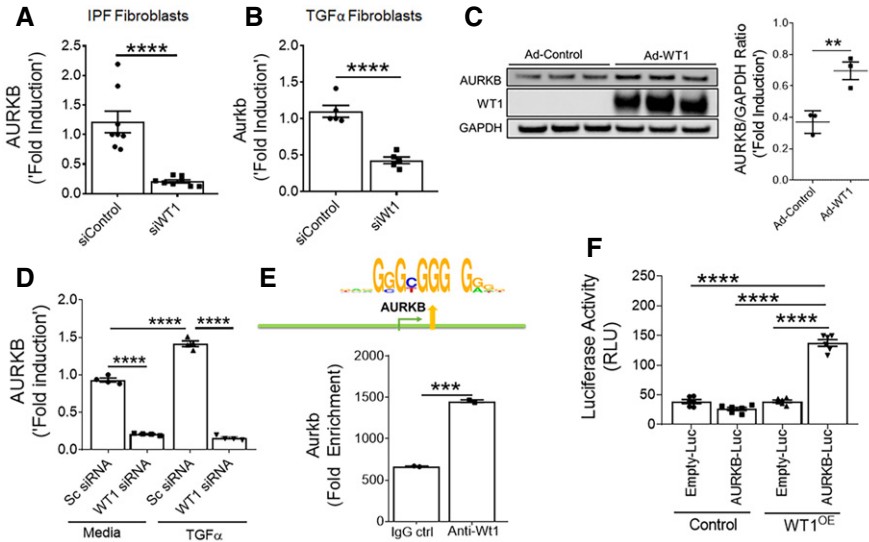

**Figure 2. WT1 regulates AURKB expression.**

A   Human IPF lung fibroblasts were transiently transfected with control or WT1 siRNA for 72 h and AURKB transcripts were quantified. ****$P < 0.00005$, unpaired $t$-test ($n = 4$).

B   Lung-resident fibroblasts from TGFα mice on Dox for 4 weeks were transiently transfected with control or WT1 siRNA for 72 h, and AURKB transcripts were quantified. ****$P < 0.00005$, unpaired $t$-test ($n = 4$).

C   Immunoblot analysis of AURKB and WT1 in the lysates of non-IPF fibroblasts transduced with control or WT1-adenoviral particles for 72 h. **$P < 0.005$, unpaired $t$-test ($n = 3$).

D   Quantification of AURKB transcripts in primary fibroblasts from IPF lung treated with either control or WT1 siRNA and stimulated with media or TGFα (100 ng/ml) for 16 h. ****$P < 0.00005$, unpaired $t$-test ($n = 4$).

E   Primary lung-resident fibroblasts were isolated from stromal cultures of TGFα mice placed on Dox for 8 weeks. Cell lysates were prepared, and the ChIP assay was performed with anti-WT1 antibody or normal rabbit IgG as a negative control using AURKB gene promoter-specific PCR primers. Non-immunoprecipitated DNA is represented as input DNA (product size, 104 bp). ***$P < 0.0005$, unpaired $t$-test ($n = 2$).

F   AURKB promoter luciferase activity was measured in HEK293 cells transfected with control or WT1 overexpressing (OE) vector. ****$P < 0.00005$, one-way-ANOVA ($n = 6$).

Data information: All data were presented as mean ± SEM. The above data were a representative of two independent experiments with similar results. $P$ values were shown in Appendix Table S6.

Source data are available online for this figure.

## AURKB functions as a positive regulator for fibroblast proliferation and survival

To gain insight into the role of AURKB in fibroblast activation, we assessed gene expression changes using next-generation sequencing in fibroblasts isolated from the lungs of TGFα mice on Dox for 10 days treated with either control- or AURKB-specific siRNA for 72 h. Knockdown of AURKB resulted in significant loss of its transcripts in resident fibroblasts (Fig EV3A). As shown in the heat map, the knockdown of AURKB has resulted in 800 upregulated and 1248 downregulated genes (≥ 1.5-fold change; $P$ value FDR ≤ 0.05) (Fig EV3B). To identify potential AURKB-driven pro-fibrotic gene transcripts in IPF, we next compared differentially expressed genes in IPF lungs (GSE53845) (DePianto et al, 2015) and AURKB-specific siRNA-treated lung fibroblasts. We identified multiple transcripts that were either upregulated (141 genes) or downregulated (50 genes) by AURKB in IPF (Fig 3A and Appendix Table S1). Functional enrichment analysis of these negatively correlated gene sets (i.e., genes upregulated in IPF lungs but downregulated in AURKB-specific siRNA-treated lung fibroblasts and vice versa) using the ToppFun application of the ToppGene Suite (Chen et al, 2009) identified several major fibroblast-specific biological processes. These

include cell proliferation, apoptosis, and ECM production (Fig 3B). Co-immunofluorescence staining was performed with anti-AURKB and anti-Ki-67 antibodies on lung sections from normal and TGFα mice on Dox for 4 weeks. We observed a notable increase in the number of AURKB and Ki-67-positive cells in the fibrotic lesions of TGFα mice on Dox for 4 weeks compared to control mice (Fig 3C). Both AURKB and Ki-67 co-localized to the same cells by immunofluorescence staining, and the number of dual positive cells for both AURKB and Ki-67 was significantly greater in the lungs of TGFα mice compared to normal mice (Fig 3D).

To test whether AURKB induces fibroproliferation, we assessed the impact of AURKB knockdown on BrdU incorporation in fibroblasts isolated from the lungs of IPF or TGFα mice on Dox for 2 weeks. We observed a significant decrease in BrdU incorporation in fibroblasts treated with AURKB-specific siRNA compared to control siRNA (Fig 3E). With the loss of AURKB in fibroblasts, we observed a significant decrease in TGFα-induced fibroproliferation (Fig 3F). To establish the mechanisms in AURKB-driven fibroproliferation, we assessed changes in AURKB-driven IPF network genes predicted to regulate fibroproliferation, such as PLK1 and CCNA2, in fibroblasts treated with AURKB-specific siRNA compared with control siRNA. Notably, the genetic knockdown of AURKB was

sufficient to attenuate the expression of AURKB-dependent genes involved in fibroproliferation (Fig 3G). These findings establish that AURKB functions as a positive regulator of fibroproliferation in IPF.

In progressive fibrosis, fibroblasts transform to myofibroblasts and develop resistance to apoptosis, thereby promoting excessive ECM deposition and the expansion of fibrotic lung lesions (Thannickal & Horowitz, 2006; Kasam *et al*, 2019). To test whether myofibroblasts express AURKB, we performed co-immunofluorescence staining of AURKB and αSMA in lung sections from control and TGFα mice at 6 weeks on Dox, when fibrosis is well established. Our co-immunostaining show that AURKB localized in the nucleus and αSMA being in cytoplasm of myofibroblasts in the subpleural fibrotic lesions of TGFα mice on Dox for 6 weeks (Fig 3H). To test whether AURKB induces fibroblast survival, the mature fibrotic lesions of IPF or TGFα mice on Dox for 6 weeks were cultured to isolate fibroblasts and apoptosis was quantified in fibroblasts treated with either control or AURKB-specific siRNA.

Notably, the loss of AURKB transcripts was sufficient to induce apoptosis in fibroblasts from IPF or TGFα mice (Fig 3I). To understand the mechanisms in AURKB-driven fibroblast survival, we quantified the expression of genes involved in apoptosis by RT–PCR. The genetic loss of AURKB was sufficient to induce the expression of pro-apoptotic gene transcripts such as Bak, Bax, and Fas in fibroblasts isolated from the lungs of IPF or TGFα mice on Dox for 6 weeks (Fig 3J and K). Together, the above findings establish that increased AURKB expression contributes to apoptosis resistance in mesenchymal cells.

### Barasertib attenuates fibroblast activation

To test whether the inhibition of AURKB activity alters fibroblasts activation, we assessed the proliferation of fibroblasts isolated from the lungs of IPF or TGFα mice on Dox for 4 weeks. Upon treatment with barasertib, a known AURKB inhibitor, we observed a

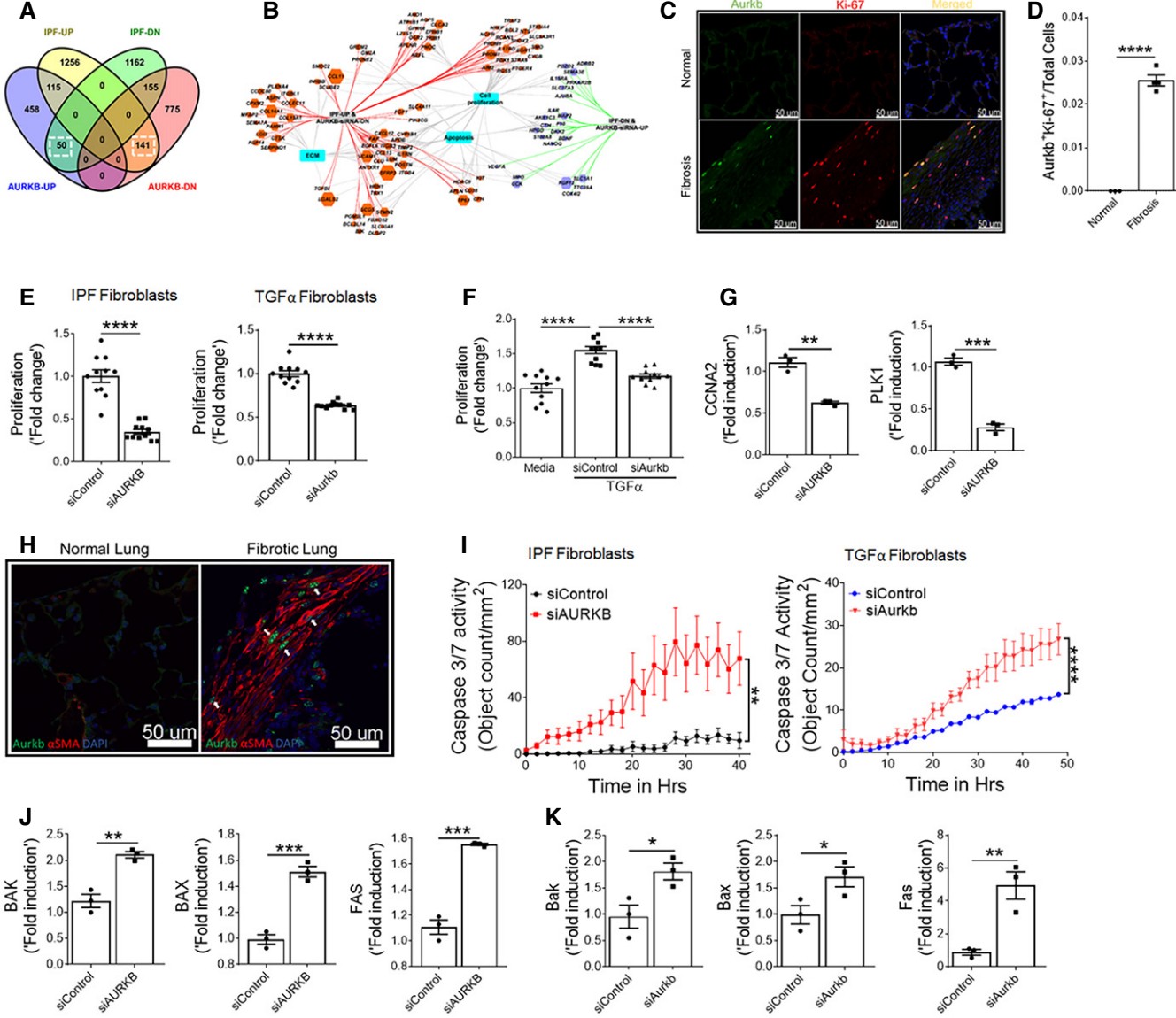

**Figure 3.**

**Figure 3.** **AURKB functions as a positive regulator of fibroproliferation and survival.**

A   Venn diagram depicting the comparison and overlap of differentially expressed genes in IPF lungs and AURKB siRNA-treated fibrotic fibroblasts. The dashed box indicates genes that were up- (141 genes) or downregulated (DN, 50 genes) in IPF lungs compared with AURKB siRNA knockdown gene expression signatures.

B   AURKB-driven genes activated in IPF were analyzed using ToppFun and visualized using Cytoscape. Red- and blue-colored circles represent genes that are up or downregulated, respectively, in IPF lungs. The turquoise-colored circle represents enriched biological processes for the inversely correlated genes between AURKB siRNA knockdown and IPF.

C   Lung sections from control and TGFα mice fed with Dox food for 4 weeks were stained for AURKB (green) and Ki-67(red). Merged image shows cells that co-express AURKB and Ki-67 (yellow).

D   Quantification of Ki-67$^+$ and AURKB$^+$ double-positive cells was performed using five confocal images per mice in control and TGFα mice. Images were obtained at 40× magnification. Scale bar: 50 µm. ****$P < 0.00005$, unpaired $t$-test ($n = 4$).

E   Proliferation was measured in primary lung-resident fibroblast isolated from either IPF lung or TGFα mice and treated with either control or AURKB siRNA. ****$P < 0.00005$, unpaired $t$-test ($n = 9$–$11$).

F   Proliferation was measured in primary lung-resident fibroblasts isolated from stromal cultures of TGFα mice on Dox for 2 weeks and transiently transfected with control or AURKB siRNA and stimulated with TGFα (20 ng/ml) for 24 h. ****$P < 0.00005$, 1-way ANOVA ($n = 9$–$11$).

G   Quantification of CCNA2 and PLK1 transcripts in human IPF fibroblasts transiently transfected with control or AURKB siRNA for 72 h. **$P < 0.005$, ***$P < 0.0005$, unpaired $t$-test ($n = 3$).

H   Lung sections from control and TGFα mice fed with Dox food for 6 weeks were stained for AURKB (green) and αSMA (red). Images were obtained at 40× magnification. Scale bar: 50 µm. ($n = 4$).

I    Quantification of apoptotic cells using Incucyte ZOOM (caspase 3/7-positive cells) in lung-resident fibroblasts isolated from IPF and TGFα mice on Dox for 6 weeks and treated with control or AURKB siRNA for 72 h. **$P < 0.005$, ****$P < 0.00005$, 2-way ANOVA ($n = 4$).

J    Quantification of Bak, Bax, and Fas gene transcripts in human IPF lung-resident fibroblasts treated with either control or AURKB siRNA for 72 h. **$P < 0.005$, ***$P < 0.0005$, unpaired $t$-test ($n = 3$).

K   Quantification of Bak, Bax, and Fas gene transcripts in lung-resident fibroblasts isolated from TGFα mice and treated with either control or AURKB siRNA for 72 h. *$P < 0.005$, **$P < 0.005$, unpaired $t$-test ($n = 3$).

Data information: All data were presented as mean ± SEM. The above data were a representative of two independent experiments with similar results. Exact $P$ values are shown in Appendix Table S6.

significant reduction in BrdU incorporation in fibroblasts isolated from IPF and TGFα model (Fig 4A and B). In support, the percent of PCNA-positive cells was significantly reduced in barasertib-treated IPF fibroblasts compared with vehicle (Fig 4C). Consistent with the loss of AURKB in fibroblasts, we observed a dose-dependent attenuation of TGFα-induced fibroblast proliferation when treated with barasertib compared to vehicle (Fig 4D). Similar to the loss of AURKB expression, inhibition with barasertib reduced the expression of AURKB-driven genes involved in fibroproliferation such as CCNA2 and PLK1 (Fig EV4). Previous studies from our group, and others, have demonstrated a pathogenic role for enhanced survival of multiple mesenchymal cells in the development and maintenance of pulmonary fibrosis (Kasam *et al*, 2019). To evaluate the effect of AURKB inhibition on apoptotic clearance of fibroblasts, we isolated fibroblasts from IPF lungs and assessed the impact of barasertib treatment on cleaved caspase 3 activity. Treatment with barasertib resulted in a significant increase in caspase 3/7 activity compared to vehicle (Fig 4E). Thus, either the loss of expression or activity of AURKB is sufficient to mitigate fibroblast activation, including fibroproliferation and survival in IPF fibroblasts and in lung fibroblasts from the TGFα mouse model.

**Barasertib attenuates TGFα-induced pulmonary fibrosis *in vivo***

To test whether treatment with barasertib prevents the development of lung fibrosis, TGFα mice were concomitantly treated with Dox to induce TGFα expression and either barasertib (40 mg/kg body weight) or vehicle twice per day for 4 weeks (Fig 5A). Induction of TGFα caused extensive subpleural, perivascular, and peribronchial fibrosis (Fig 5B). TGFα-activated mice that were concomitantly given barasertib showed reduced collagen staining with infrequent scattered small fibrotic areas. Further, we observed a significant decrease in the lung weights and hydroxyproline levels in TGFα

mice treated with barasertib compared to vehicle-treated mice (Fig 5C and D). In agreement with the transcriptional changes, we observed a significant decrease in lung collagen levels in TGFα mice treated with barasertib compared to vehicle-treated TGFα mice on Dox for 4 weeks (Fig 5C). Further, gene expression analysis by RT–PCR demonstrated a reduction in the expression of ECM genes such as Col1α, Col5α, and Fn1 in barasertib-treated TGFα mice compared to vehicle-treated TGFα mice (Fig 5E). Together, these findings suggest that inhibition of AURKB activity by barasertib attenuates progressive fibrotic changes in TGFα mice.

To determine whether the anti-fibrotic effects of barasertib *in vivo* are due to decreased proliferation in fibrotic lung lesions, we next immunostained the lung sections with Ki-67 and found a significant increase in immunoreactivity for Ki-67 in the subpleural and adventitial fibrotic lesions of TGFα mice treated with vehicle compared to non-fibrosis control mice (Fig 6A). Notably, TGFα mice treated with barasertib displayed a significant reduction in Ki-67 staining in both subpleural and adventitial fibrotic lesions. Additionally, immunoblot analysis in lung lysates demonstrated decreased PCNA levels in the barasertib-treated TGFα mice compared to vehicle-treated TGFα mice (Fig 6B). The transcript levels for genes implicated in cell proliferation, including Aurkb, were significantly reduced in barasertib-treated TGFα mice lungs compared to vehicle-treated mice (Fig 6C). Also, we observed a significant reduction in the expression of Wt1 in the lungs of TGFα mice treated with barasertib compared to vehicle-treated TGFα mice (Fig EV5).

To determine whether AURKB inhibition influences the progression of established and ongoing pulmonary fibrosis following 3 weeks of Dox treatment when fibrosis is already manifest, TGFα mice were administered barasertib while remaining on Dox for an additional 3 weeks (Fig 7A). The right lung weights were increased in TGFα mice compared to control mice on Dox for 6 weeks. This increase in lung weights was attenuated in TGFα mice treated with

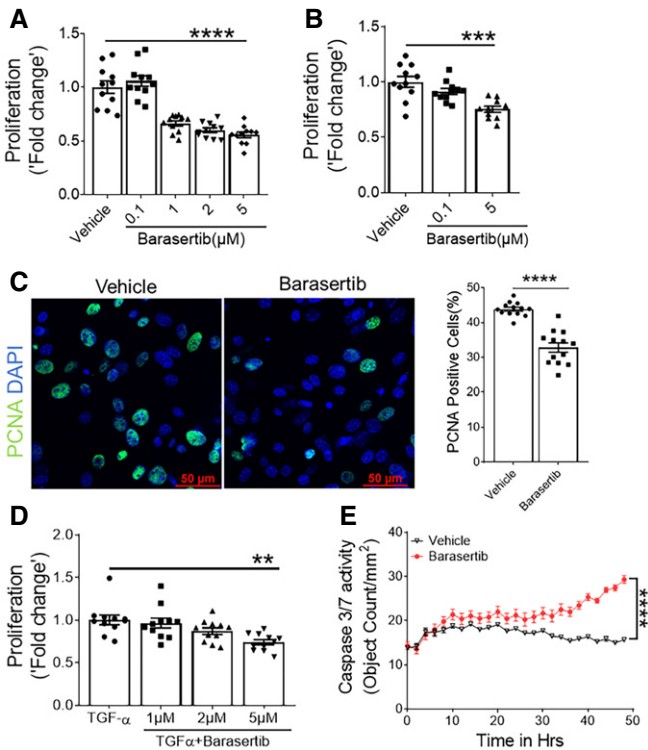

**Figure 4. Blockade of AURKB activity impacts fibroblast proliferation and survival.**

A Proliferation was assessed using BrdU incorporation assay in human IPF fibroblasts treated with indicated doses of barasertib for total of 48 h. ****$P$ < 0.00005, 1-way ANOVA ($n$ = 9–11).

B Proliferation was assessed using BrdU incorporation assay in fibroblasts isolated from TGFα mice lung and treated with indicated doses of barasertib for total of 48 h. ***$P$ < 0.0005, 1-way ANOVA ($n$ = 9–11).

C Primary lung-resident fibroblasts isolated from TGFα mice on Dox for 4 weeks were treated with vehicle or 5 μM barasertib for 48 h and immunostained using PCNA antibody. Images were obtained at 40× magnification. Scale bar: 50 μm. The number of PCNA-positive cells and total DAPI-positive cells was quantified using MetaMorph image analysis software. Proliferation is indicated as the percentage of proliferating cells in total DAPI-positive cells. ****$P$ < 0.00005, unpaired $t$-test ($n$ = 3–4).

D Proliferation was measured in primary fibroblasts treated with TGFα (20 ng/ml) and indicated doses of barasertib for 48 h. **$P$ < 0.005, 1-way ANOVA ($n$ = 9–11).

E Quantification of apoptotic cells using Incucyte ZOOM (caspase 3/7-positive cells) in resident fibroblasts isolated from IPF lung stromal cultures and treated with vehicle or 5 μM Barasertib. ****$P$ < 0.00005, 2-way ANOVA ($n$ = 4).

Data information: All data were presented as mean ± SEM. The above data were a representative of two independent experiments with similar results. $P$ values were shown in Appendix Table S6.
Source data are available online for this figure.

barasertib when compared to vehicle-treated TGFα mice (Fig 7B). Western blot analysis revealed a significant decrease in Col1α and Fn1 protein levels in barasertib-treated TGFα mice lungs compared to vehicle-treated (Fig 7C). TGFα mice on Dox for 6 weeks develop severe fibrotic lung disease including subpleural and adventitial thickening and the loss of lung function. Intervention therapy with barasertib resulted in a marked reduction in subpleural and adventitial fibrosis as evidenced by masons trichrome staining (Fig 7D).

Notably, barasertib treatment resulted in improvements in resistance and elastance and prevented a decrease in lung compliance during TGFα-induced pulmonary fibrosis (Fig 7E). To assess possible effects of doxycycline and DMSO treatments individually on inflammation and fibrosis, CCSP-rtTA and TetO-TGFα mice were treated with Dox-containing food for 6 weeks or vehicle containing 5% DMSO for 3 weeks. We observed no significant effect of doxycycline or DMSO treatments on inflammation, lung collagen deposition, and lung function compared to untreated littermate control mice (Appendix Fig S1).

**Barasertib attenuates bleomycin-induced pulmonary fibrosis**
***in vivo***

To substantiate our hypothesis that AURKB inhibition with barasertib attenuates pulmonary fibrosis, we used an alternative mouse model of bleomycin-induced pulmonary fibrosis (Singh *et al*, 2017; Sontake *et al*, 2018). Pulmonary fibrosis was induced in wild-type mice by treating with bleomycin intradermally 5 days per week for 4 weeks, while the last 2 weeks they received either vehicle or barasertib (Fig 8A). Lung sections stained with Masson's trichrome demonstrated significant increase in collagen staining in the lung parenchyma of bleomycin-treated mice compared with vehicle-treated control mice. This increase in Masson's trichrome staining was attenuated with barasertib treatment compared to vehicle-treated mice during bleomycin-induced pulmonary fibrosis (Fig 8B). Similarly, barasertib treatment resulted in a significant decrease in the lung hydroxyproline levels compared to vehicle-treated fibrotic mice (Fig 8C). To determine whether barasertib treatment has any negative effect on AURKB-regulated gene network, we analyzed total lung transcripts using RT–PCR. In particular, transcripts of Col1α, Col3α, Col5α, Col14α, Col15α, Ccna2, Ccnb1, and Fas were significantly decreased in barasertib-treated mice compared to vehicle-treated fibrotic mice (Fig 8D). Barasertib treatment also caused a significant decrease in the αSMA protein levels in the lung lysates of barasertib-treated mice compared to vehicle-treated fibrotic mice (Fig 8E). Taken together, our findings provide complementary *in vivo* evidence that barasertib treatment attenuates collagen deposition and plays a protective role in pulmonary fibrosis.

## Discussion

We report a mitotic kinase called aurora kinase B that links fibroblast activation with fibrotic lung remodeling in IPF. AURKB is a member of the aurora kinases family and a mitotic kinase that plays a critical role in multiple steps of the cell cycle, including cytokinesis (Willems *et al*, 2018). AURKB is overexpressed in a wide range of tumor types and has been considered a marker of prognosis (Portella *et al*, 2011; Tang *et al*, 2017; Chieffi, 2018). Our results demonstrate that AURKB is upregulated in fibroblasts isolated from fibrotic lung lesions of IPF and a mouse model of TGFα-induced pulmonary fibrosis. Our immunostainings suggest upregulation of AURKB in a subset of mesenchymal cells in subpleural fibrotic lesions and fibrotic foci of IPF lungs. Co-immunostainings with Ki67 and αSMA show that AURKB is upregulated in both proliferating fibroblasts and myofibroblasts that populate the fibrotic lung lesions.

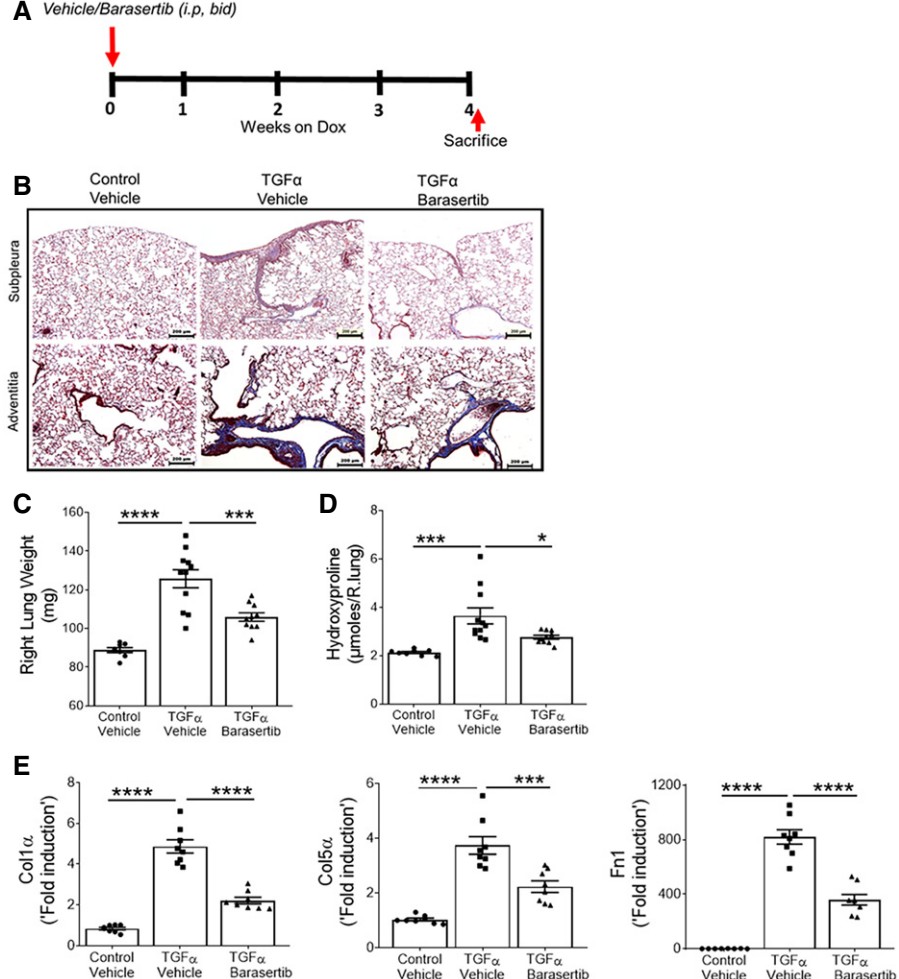

**Figure 5.** *In vivo* **Barasertib treatment prevents from TGFα induced lung fibrosis.**

A   Schematic illustration of barasertib preventive treatment protocol. Control and TGFα mice were treated with either vehicle or barasertib (40 mg/kg; twice a day) for 4 weeks, while they were fed with Dox-containing food.

B   Representative images of Masson's trichrome-stained lung sections from the vehicle- and barasertib-treated mice. Images were obtained at 10× magnification. Scale bar: 200 μm.

C   Quantification of right lung weight of mice treated with vehicle or barasertib. ***P < 0.0005, ****P < 0.00005, 1-way ANOVA (n = 8–10 mice/group).

D   Quantification of total lung hydroxyproline levels in mice treated with vehicle or barasertib. *P < 0.05, ***P < 0.0005, 1-way ANOVA (n = 8–10 mice/group).

E   Quantification of Col1α, Col5α, and Fn1 gene transcripts in total lung of mice treated with vehicle or barasertib. ***P < 0.0005, ****P < 0.00005, 1-way ANOVA (n = 8 mice/group).

Data information: All data were presented as mean ± SEM. *P* values were shown in Appendix Table S6.

Using knowdown studies, we identified AURKB-driven gene networks that were fibroblast-specific and dysregulated in IPF, such as genes involved fibroproliferation, survival, and ECM production. In support of its pro-fibrotic actions, loss of AURKB resulted in attenuation of fibroproliferation and increased apoptotic clearance of fibroblasts, suggesting that AURKB is a positive regulator of fibroblast activation. Furthermore, AURKB is known to induce proliferation and survival in cancer cells during tumor development. Elevated expression of AURKB in thyroid anaplastic carcinoma cells can induce higher growth rate, and inhibition of AURKB expression by RNA interference has resulted in reduced proliferation of these cells (Sorrentino *et al*, 2005). Similarly, the knockdown of Epstein–Barr nuclear antigen 3C (EBNA3C)-induced AURKB expression resulted in increased activity of caspases 3 and 9 along with poly (ADP-ribose) polymerase 1 (PARP1) cleavage (Jha *et al*, 2013). Inhibition of AURKB activity by barasertib attenuated fibroproliferation and induces the apoptotic clearance in fibroblasts of IPF and TGFα mice. We confirmed that pharmacological inhibition of AURKB activity or siRNA-mediated inhibition of AURKB expression is sufficient to inhibit the expression of genes such as Plk1 and CcnA2 implicated in fibroblast proliferation. With AURKB inhibition, we observed a significant increase in the expression of pro-apoptotic genes such as Bak, Bax, and Fas, which are downstream effectors in caspase-mediated apoptotic clearance. This agrees with our recent report that apoptosis-resistant fibroblasts isolated from fibrotic lung express low levels of pro-apoptotic genes such as Bak and Bax

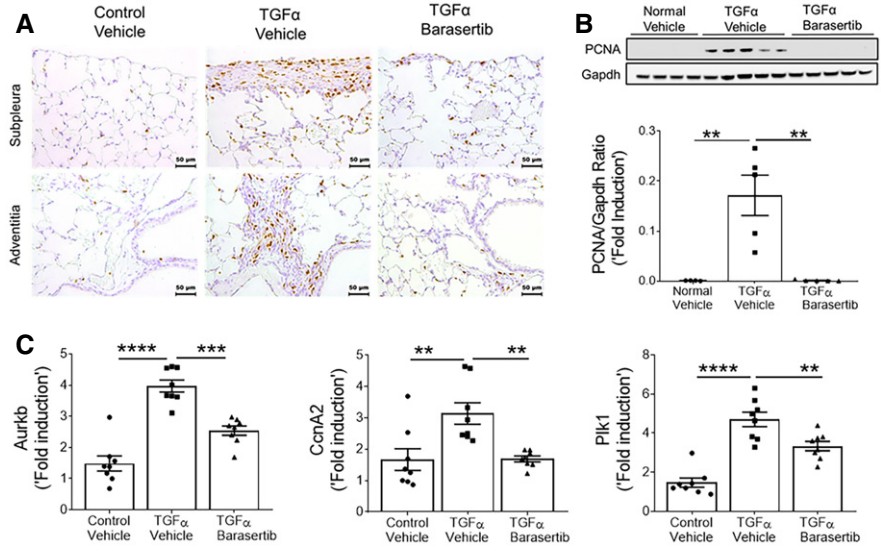

**Figure 6.** *In vivo* **barasertib treatment attenuates mesenchymal proliferation.**

A  Lung sections from vehicle- and barasertib-treated mice were immunostained using Ki-67 antibody. Representative images were obtained at 20× magnification. Scale bar: 50 μm.

B  Immunoblot analysis in lung lysates from vehicle- and barasertib-treated mice using PCNA antibody. Gapdh is used as loading control. **$P < 0.005$, 1-way ANOVA ($n = 4$–$5$ mice/group).

C  Quantification of Aurkb, Plk1, and CcnA2 gene transcripts in total lungs of mice treated with vehicle or barasertib. **$P < 0.005$, ***$P < 0.0005$, ****$P < 0.00005$, 1-way ANOVA ($n = 8$ mice/group).

Data information: All data were presented as mean ± SEM. *P* values were shown in Appendix Table S6.

Source data are available online for this figure.

compared to normal fibroblasts (Kasam *et al*, 2019). Similarly, a recent publication from Bogen *et al* showed that barasertib induces apoptosis in multiple types of MYCN-amplified neuroblastoma cell lines (Bogen *et al*, 2015). This phenomenon was also documented in multiple cancer types, including breast cancer, colorectal cancer, melanoma, and leukemia (Gully *et al*, 2010; Alferez *et al*, 2012; Yamauchi *et al*, 2013; Porcelli *et al*, 2015).

Our previous work demonstrates a role for WT1 in fibroblast activation, including fibroproliferation, myofibroblast transformation, and ECM gene expression in the pathogenesis of IPF (Sontake *et al*, 2015, 2018). This work expands upon these findings, demonstrating that WT1 upregulates AURKB expression in fibroblasts. The fibroproliferative changes detected in TGFα mice may be the result of WT1-driven AURKB expression. ChIP analysis and the use of an AURKB promoter-driven luciferase assay revealed that WT1 directly interacts with the promoter elements of AURKB to drive its expression in fibroblasts. Our previous study demonstrates that WT1 regulates the myofibroblast transformation by interacting with αSMA promoter region and promoting its expression (Sontake *et al*, 2018). Hence, we speculate that WT1 promotes the fibroblasts activation at least in part by regulating the expression of AURKB. These findings posit a new role for WT1-AURKB axis in fibroproliferation. However, future studies are needed to evaluate possible binding partners and other molecular regulators downstream of WT1 and AURKB in fibroblast activation and pulmonary fibrosis.

These findings have therapeutic implications as they suggest targeting AURKB activity inhibits fibroblast activation, particularly when multiple pro-fibrotic growth factors promoting alternative

fibroproliferative pathways are present. We evaluated the *in vivo* effects of AURKB inhibition by barasertib using mouse models of TGFα- and bleomycin-induced pulmonary fibrosis. Published studies have shown elevated levels of TGFα in the lung lavage fluid and fibrotic lung lesions of IPF patients compared to healthy controls (Madtes *et al*, 1988; Baughman *et al*, 1999). In support, overexpression of TGFα in mice has resulted in the development of pathologic fibrotic lesions similar to those found in IPF, including fibroproliferation, resistance to apoptosis, and subpleural fibrosis migrating into the interstitium, myofibroblast transformation, and a progressive decline in lung function (Hardie *et al*, 2004, 2010; Sontake *et al*, 2018; Kasam *et al*, 2019). Moreover, gene expression profiles after expression of TGFα were more similar to IPF samples compared to bleomycin model (Hardie *et al*, 2007; Kasam *et al*, 2019). In a recent study, we demonstrated that the FDA-approved anti-fibrotic agent nintedanib inhibited TGFα-induced pulmonary fibrosis by limiting both fibroproliferation and myofibroblast survival (Kasam *et al*, 2019). Similarly, in the TGFα model used in the current study, *in vivo* treatment with barasertib either as a preventive or as a therapeutic strategy led to decreased fibroproliferation and myofibroblast accumulation in the pathogenesis of pulmonary fibrosis. Our lung physiology measurements using FlexiVent have demonstrated that treatment of mice with barasertib improved lung function during TGFα-induced pulmonary fibrosis. These physiological changes are in agreement with observed changes in the expression of genes associated with proliferation, apoptosis, and ECM. Furthermore, we demonstrated anti-fibrotic effects of barasertib using a chronic mouse model of bleomycin-induced pulmonary fibrosis (Madala

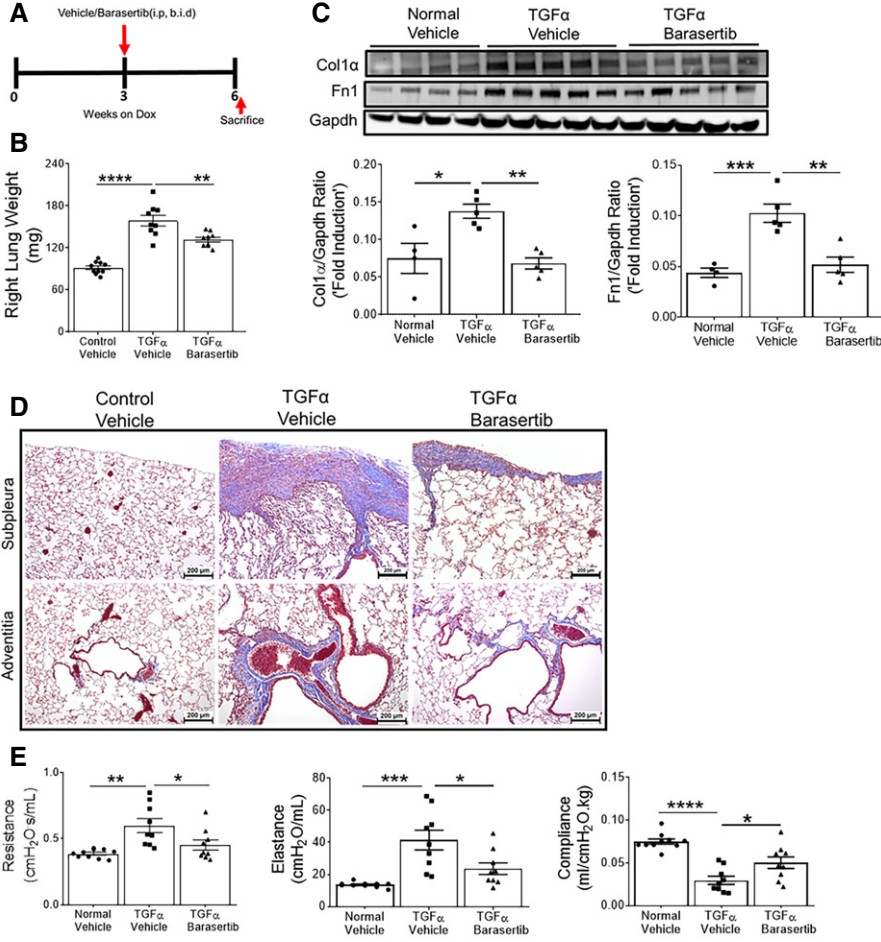

**Figure 7. Therapeutic barasertib treatment reduces established and ongoing lung fibrosis.**

A Schematic illustration of barasertib treatment protocol. Control and TGFα mice were treated with either vehicle or barasertib (40 mg/kg; twice a day) for last 3 weeks, while they were fed with Dox-containing food for total of 6 weeks.

B Quantification of right lung weight of mice treated with vehicle and barasertib. **$P < 0.005$, ****$P < 0.00005$, 1-way ANOVA ($n = 9–10$ mice/group).

C Western blot analysis in lung lysates from vehicle- and barasertib-treated mice using Col1α and Fn1 antibodies. Gapdh is used as loading control. *$P < 0.05$, **$P < 0.005$, ***$P < 0.0005$, 1-way ANOVA ($n = 4–5$ mice/group).

D Representative images of Masson's trichrome-stained lung section from mice treated with vehicle and barasertib. Images were obtained at 10× magnification. Scale bar: 200 μm.

E Quantification of lung mechanics in mice treated with vehicle and barasertib. *$P < 0.05$, **$P < 0.005$, ***$P < 0.0005$, ****$P < 0.00005$, 1-way ANOVA ($n = 9–10$ mice/group).

Data information: All data were presented as mean ± SEM. *P* values were shown in Appendix Table S6.

*et al*, 2016a; Singh *et al*, 2017; Sontake *et al*, 2018). A large body of evidence supports the concept that macrophages, fibrocytes, and other immune cells secrete paracrine factors to cause excessive fibroproliferation (Madala *et al*, 2014; Desai *et al*, 2018). Importantly, the frequency of proliferating fibroblasts was elevated in the expanding areas of fibrotic foci and infusion of these proliferating mesenchymal progenitors resulted in the progressive fibrosis in mice (Xia *et al*, 2017). However, the expansion of fibrotic lesions is also attributed to the resistance to programmed cell death in myofibroblasts that accumulate in the mature fibrotic lesions (Horowitz & Thannickal, 2019; Hinz & Lagares, 2020). In support, myofibroblasts isolated from IPF lungs displayed abnormalities in apoptotic pathways, senescence, and defective autophagy (Patel

*et al*, 2012; Alvarez *et al*, 2017; Kasam *et al*, 2019). Also, it is now increasingly recognized that the pathogenesis of IPF is complex likely in part due to the molecular and functional heterogeneity of fibroblasts and other lung cells (Rock *et al*, 2011; Reyfman *et al*, 2018; Tsukui *et al*, 2020). Therefore, use of inhibitors that block both proliferation and myofibroblast survival could be more beneficial to attenuate established and ongoing pulmonary fibrosis. Barasertib is currently in clinical studies, and it has been evaluated in patients with solid malignant tumors and myeloid leukemia (Lowenberg *et al*, 2011). Moreover, barasertib treatment in murine xenograft model has potentiated the anti-proliferative properties of vincristine and daunorubicin (Yang *et al*, 2007). In human xenograft models, administration of barasertib was effective against various

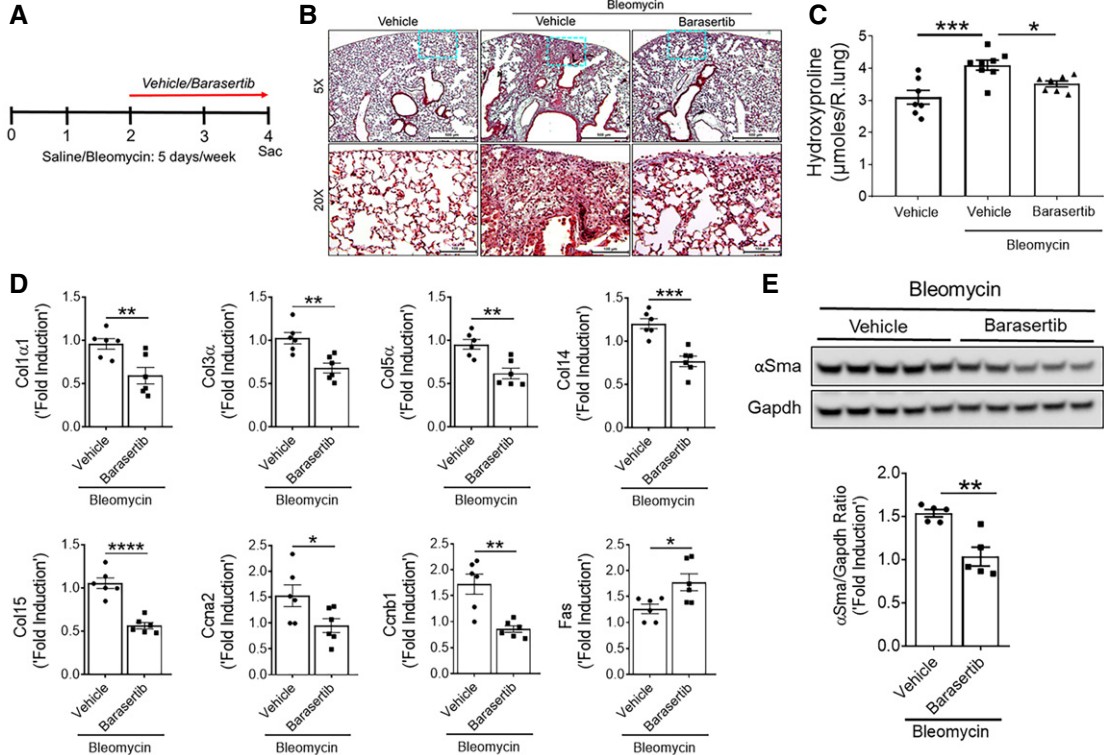

**Figure 8. Therapeutic barasertib treatment attenuates bleomycin-induced lung fibrosis.**

A   Schematic illustration of barasertib treatment protocol. Mice were treated with either vehicle or barasertib (40 mg/kg; twice a day) for last 2 weeks, while they were injected intradermally with either saline or bleomycin 5 days per week for total of 4 weeks.

B   Representative images of Masson's trichrome-stained lung section from mice treated with vehicle and barasertib. Images were obtained at 10× magnification. Scale bar: 200 μm.

C   Quantification of total lung hydroxyproline in mice treated with vehicle and barasertib. *$P < 0.05$, ***$P < 0.0005$, 1-way ANOVA ($n = 7$–8 mice/group).

D   Quantification of Col1α1, Col3α, Col5α, Col14, Col15, αSma, Ccna2, and Fas gene transcripts in total lungs of mice treated with vehicle or barasertib. *$P < 0.05$, **$P < 0.005$ ***$P < 0.0005$, ****$P < 0.00005$, 1-way ANOVA ($n = 6$ mice/group).

E   Quantification of αSma protein levels in lung lysates from vehicle- and barasertib-treated mice. Gapdh is used as loading control. Data are presented as mean ± SEM. **$P < 0.005$, unpaired *t*-test ($n = 5$ mice/group).

Data information: All data were presented as mean ± SEM. $P$ values were shown in Appendix Table S6.

---

tumor types. Barasertib was shown to inhibit the growth of human colon, lung, and hematologic tumor xenografts by reducing cell division and inducing apoptosis-mediated cell death (Wilkinson *et al*, 2007). Barasertib was tested in clinical trials for patients with acute myeloid leukemia (AML) and various other tumors (Tsuboi *et al*, 2011; Dennis *et al*, 2012; Collins *et al*, 2015). The anti-tumor effects of barasertib were enhanced when used in combination with dexamethasone (Evans *et al*, 2008). Overall, these results indicate that fibroblast-specific expression of AURKB is a critical driver of fibroblast activation and that therapeutic targeting of AURKB may serve as an ideal therapy in IPF.

In summary, we have identified and described the pathological role for AURKB in fibroblast activation and pulmonary fibrosis. We identified a role for WT1 in AURKB expression and AURKB-driven transcripts involved in proliferation and survival of fibroblasts in the pathogenesis of pulmonary fibrosis. Our preclinical findings support the use of barasertib as an anti-fibrotic agent to attenuate fibroblast activation involved in established and ongoing pulmonary fibrosis.

## Materials and Methods

### Mouse models of TGFα and bleomycin-induced pulmonary fibrosis

Clara cell-specific promoter (CCSP)-driven rtTA mice (CCSP-rtTA) and TGFα overexpressing mice were generated as described previously (Hardie *et al*, 2004). Hemizygous (TetO)$_7$-cmv TGFα mice were crossed to CCSP-rtTA mice to generate bitransgenic mice (CCSP-rtTA;TGFα$^{tg}$) and littermate controls (CCSP-rtTA/−). Both male and female gender mice at age 10–16 weeks were used for all the experiments. Spatiotemporal overexpression of TGFα was induced by feeding the mice with doxycycline (62.5 mg/kg) containing chow. Stock solutions of barasertib (AZD-1152-HQPA; Selleckchem, TX, USA) were prepared freshly in vehicle (5% DMSO and 50% PEG-300). Previous pharmacokinetic studies have indicated that sustained and efficient AURKB inhibition could be achieved with barasertib administered at a dose range of 25–50 mg/kg every 8–12 h in mice (Marchetti *et al*, 2013; Floc'h *et al*, 2017). Therefore,

we treated mice with vehicle or barasertib (40 mg/kg body weight) by intraperitoneal injections twice a day with 12-h intervals. For *in vivo* intervention studies, all groups of mice were started on Dox for total 6 weeks. At the beginning of week 3, mice were treated with either vehicle or barasertib for the final 3 weeks of the study (Sontake *et al*, 2017). For bleomycin-induced mouse model, 10- to 14-week-old C57BL/6J mice (JAX Stock # 000664) were treated intradermally with saline or bleomycin (6 U/kg body weight) 5 days per week for 4 weeks intradermally as described (Singh *et al*, 2017). For *in vivo* intervention studies, all mice were treated with saline or bleomycin for the total 4 weeks, and at the beginning of week 2, mice were treated with either vehicle or barasertib for the final 2 weeks of the study. All the mice were housed under specific pathogen-free conditions, and all animal experiments were performed under protocols approved by the Institutional Animal Care and Use Committee of the Cincinnati Children's Hospital Research Foundation.

## Human samples

De-identified human IPF and control non-IPF lung tissues were obtained from the Interstitial Lung Disease Biorepository at the University of Michigan Medical School and provided by Steven Huang in the Department of Internal Medicine. Specimens consisted of paraffin-embedded and freshly isolated lung tissue of patients who underwent lung transplantation for IPF. IPF was diagnosed according to the American Thoracic Society consensus criteria (Raghu *et al*, 2011). Lung samples from donors with no lung disease were used as normal lung biopsies. All tissues were acquired using research protocols and informed consent and approved by the Michigan Medicine Institutional Review Board (HUM00105694). The experiments conformed to the principles set out in the WMA Declaration of Helsinki (https://www.wma.net/policies-post/wma-declaration-of-helsinki-ethical-principles-for-medical-research-involving-human-subjects/) and the Department of Health and Human Services Belmont Report (http://www.hhs.gov/ohrp/humansubjects/guidance/belmont.html).

## RNA isolation and RT–PCR

Total RNA from primary cells and lung tissue was prepared using RNeasy kit (Qiagen Sciences, Valencia, CA) as described previously (Kasam *et al*, 2019). Real-time PCR was performed using the CFX384 Touch Real-Time PCR detection system and SYBR select master mix (Bio-Rad, Hercules, CA). Target gene transcripts in each sample were normalized to mouse hypoxanthine guanine phosphoribosyltransferase (Hprt) or human beta-actin or Gapdh. All the real-time primer details are provided in Appendix Tables S2 and S3.

## Western blot

Western blot analysis was performed as described earlier (Madala *et al*, 2016a). Briefly, cell lysate was prepared either from primary cells or from lung tissue using the RIPA lysis buffer containing protease and phosphatase inhibitors. The protein concentrations in the supernatants were quantified using BCA kit (Thermo Fisher Scientific, Waltham, MA). After SDS–PAGE separation, proteins were transferred to nitrocellulose membrane and then were probed using specific primary antibodies followed by detection with HRP-conjugated secondary antibodies. Bands were quantified using the volume integration function of the phosphor imager software, Multi-gage (Fujifilm, Valhalla, NY). Details of antibodies used in the Western blot are provided in Appendix Table S5.

## Histology and immunohistochemistry

Histology and immunostaining were performed as described previously (Sontake *et al*, 2017). Briefly, lungs were inflated and fixed with formalin. Paraffin-embedded lungs were cut into 5-μM sections and stained for masons trichrome. For immunostaining, lung sections were deparaffinized, antigen retrieval was performed using 10 mM citric acid (pH 6.0), and sections were incubated in blocking solution before they were incubated with specific primary antibody overnight. DAB-stained sections were mounted and visualized using a bright field microscope (Leica Microsystems).

## Immunofluorescence and confocal imaging

Immunofluorescence, confocal imaging, and cell quantification were performed as described previously (Madala *et al*, 2014). Briefly, OCT medium-embedded lung sections were immunostained using specific primary antibodies and followed by detection with appropriate secondary antibodies conjugated with Alexa Fluor-488 or 568 or 647. For immunocytochemistry, primary cells were fixed with methanol or 4% paraformaldehyde and were blocked using BSA solution before they were incubated with primary antibodies. Lung sections or primary cells were stained with DAPI for nuclear staining. Confocal images were collected using a Nikon AIR-A1 laser scanning confocal microscope. Imaris (version 7.2.0; Bit plane) was used for image analysis and automated cell quantification. Details of the antibodies used for immunostaining are provided in Appendix Table S4.

## Quantification of apoptotic death

Apoptotic cell death was measured as described previously using IncuCyte ZOOM real-time imaging system (Kasam *et al*, 2019). Briefly, primary cells were incubated in Caspase 3/7 Green apoptosis assay reagent (Caspase 3/7 substrate conjugated to green fluorophore; Essence Bioscience) at a final concentration of 5 μM/ml. Time-lapse fluorescence imaging was performed using the IncuCyte ZOOM system (Essen BioScience); 9 images per well at 20× magnification were collected every 2 h for 24–48 h. The average number of green objects produced by the apoptotic cells was measured using Incucyte ZOOM software 2015A.

## BrdU cell proliferation assay

Cell proliferation was quantified using BrdU kit (Cell signaling technology, Denver, CO) as described previously (Sontake *et al*, 2017). Briefly, primary lung-resident fibroblasts that were either transiently transfected with Aurkb siRNA or treated with vehicle or barasertib were incubated with BrdU labeling solution for 24 h. The cells were then fixed 24 h after BrdU labeling, and immunodetection of BrdU was performed according to the manufacturer's protocol.

### siRNA-mediated knockdown

Stealth siRNA-mediated transfection was performed in human or mouse lung-resident fibroblasts using Lipofectamine 3000 transfection kit as described previously (Sontake *et al*, 2015). Stealth negative control, Aurkb stealth siRNA, and WT1 stealth siRNA were purchased from Invitrogen: Human Aurora kinase B (Cat# HSS190048), mouse Aurora kinase B (Cat# MSS209696), human WT1 (Cat# HSS111390), mouse WT1 (Cat# MSS212628), and stealth negative control siRNA (Cat# 12935300).

### Adenoviral transduction

Human lung-resident fibroblasts purified from lung stromal cultures using CD45-microbeads by negative selection were transduced with control or WT1-overexpressing adenoviral particles (000488A, Abm, Richmond, Canada) as described previously (Sontake *et al*, 2018). Cells were harvested 48–72 h post-viral transduction.

### Human and mouse lung primary mesenchymal cell culture

Human and mouse lung mesenchymal cell cultures were prepared as described previously (Sontake *et al*, 2018). Human and mouse mesenchymal cells were cultured in DMEM with 10% FBS and IMDM with 5% FBS media, respectively. Primary cells used in the experiments were between passages 1 and 5.

### Hydroxyproline measurement and lung mechanics

Hydroxyproline was assessed by a colorimetric assay, as previously described (Madala *et al*, 2016b). Lung function parameters such as resistance, elastance, and compliance were measured using a computerized Flexi Vent system (SCIREQ, Montreal, Canada) as described previously (Singh *et al*, 2017).

### RNA-sequencing and computational analysis

RNA was prepared from lung-resident fibroblasts from TGFα mice on Dox for 10 days and transfected with control or Aurkb siRNA. RNA-sequencing and differential gene expression analysis were performed as described previously (Sontake *et al*, 2018). Aurkb siRNA-generated gene signature was queried against already available IPF gene signature (GSE53845) (DePianto *et al*, 2015). Functional enrichment analysis of the negatively correlated gene sets between IPF lungs and Aurkb siRNA-treated fibroblasts was analyzed using the ToppFun application of the ToppGene Suite (Chen *et al*, 2009).

### ChIP-seq PCR assay

ChIP-seq PCR was performed using the ChIP assay kit (Cell signaling technology, Denver, CO) as described previously (Sontake *et al*, 2018). Briefly, lung-resident fibroblasts were purified from lung stromal culture of CCSP/TGFα mice on Dox-containing food for 8 weeks. Cells were cross-linked with 1% formaldehyde, and immunoprecipitation was performed with anti-WT1 (catalog 12609-1-AP, Proteintech, Rosemont, IL) or isotype control IgG (Cell Signaling Technology, Denver, CO) at 4°C as described in the

## The paper explained

### Problem
Idiopathic pulmonary fibrosis (IPF) is a fatal fibrotic lung disease caused by activation of multiple pro-fibrotic pathways in fibroblasts to induce excessive proliferation, survival, and ECM secretion. Hence, understanding converging points in fibroblast activation is essential to develop novel and effective therapies against IPF.

### Results
Aurora kinase B is a key mitotic kinase that we found upregulated in lung fibroblasts of IPF and mouse models of TGFα- and bleomycin-induced pulmonary fibrosis. This increase in AURKB expression in lung-resident fibroblasts is driven by multiple pro-fibrotic growth factors and a transcription factor called Wilms tumor 1. Using both *in vitro* and *in vivo* models of pulmonary fibrosis, we demonstrate that AURKB upregulation is involved in fibroblast activation and the pharmacologic inhibition of AURKB attenuates ongoing pulmonary fibrosis in mouse models of TGFα- and bleomycin-induced pulmonary fibrosis.

### Impact
This study provides the first evidence that the AURKB functions as a positive regulator for fibroproliferation and survival in the progressive expansion of fibrotic lesions in IPF. Our new findings support the mechanism of the WT1-AURKB axis to induce fibroproliferation in IPF. Barasertib treatment is an effective therapy to attenuate ongoing fibrosis in mouse models of severe fibrotic lung disease and may prove useful as a potential therapeutic option for IPF.

manufacturer's protocol. DNA was purified, and ChIP-qPCR was performed using the CFX384 Touch Real-Time PCR detection system and SYBR select master mix (Bio-Rad, Hercules, and CA).

### Luciferase assay

HEK293 cells were transiently co-transfected with vectors containing empty promoter or AURKB promoter fused with luciferase reporter (SwitchGear Genomics, Menlo Park, CA) and pLJM1 alone or with pLJM1 vector expressing WT1 using lipofectamine-3000 transfection reagent (Invitrogen). After 72 h, cells were lysed, and Renilla luciferase activity was measured using LightSwitch Luciferase Assay Reagent (SwitchGear Genomics) in a Flex Station 3 microplate reader (Molecular Devices). Luciferase activity is reported as relative luminescence units (RLU).

### Statistical analysis

For *in vivo* studies, we used the sample size of seven to ten per group based on our published studies using preclinical mouse model of pulmonary fibrosis. We considered sex as a biological variable and randomized both male and females animals in groups to analyze animal data in aggregate. Also, the details of experimental groups were blinded to investigators for biochemical and histological evaluations. All data were analyzed for statistical significance using Prism (version 8; Graph Pad). We used *t*-test to compar between two groups and one-way ANOVA with Turkey's multiple comparison post-test to compare between more than two experimental groups. All data were expressed as the mean ± SEM.

Data were considered statistically significant for $P < 0.05$. The exact $P$ values for each figure can be found in Appendix Table S6.

## Data availability

The RNA-seq datasets produced in this study are available at Gene Expression Omnibus (GEO) under the accession number GSE153898 (https://www.ncbi.nlm.nih.gov/geo/query/acc.cgi?acc = GSE153898).

Expanded View for this article is available online.

## Acknowledgements

We are thankful for the veterinary services and research pathology core at Cincinnati Children's Hospital Medical Center for help in the study. This study was supported in part by the NIH 1R01 HL134801 (SKM), 1R21 AG059533 (SKM) and the US Department of Defense funds, W81XWH-17-1-0666 (SKM), and NIH NCATS grant 1UG3TR002612 (AGJ).

## Author contributions

RKK and SKM conceived the project, designed experiments, analyzed data and wrote the manuscript; RKK performed most of the experiments; DS performed immunostainings; AGJ and SG performed bioinformatics analysis; SKH provided human lung tissues and edited manuscript; GBR and AGJ analyzed data and edited manuscript.

## Conflict of interest

The authors declare that they have no conflict of interest.

## For more information

(i)      Pulmonary Fibrosis Foundation: https://www.pulmonaryfibrosis.org/

(ii)     American Thoracic Society: https://www.thoracic.org/patients/patient-resources/resources/idiopathic-pulmonary-fibrosis.pdf

(iii)    Dr. Madala's webpage: https://www.cincinnatichildrens.org/bio/m/satish-madala

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
