## [Review Process File · EMBO Molecular Medicine]

Inhibition of Aurora Kinase B attenuates fibroblast activation and pulmonary fibrosis

Rajesh Kasam, Sudhir Ghandikota, Divyalakshmi Soundararajan, G Bhanuprakash Reddy, Steven Huang, Anil Jegga, and Satish Madala

DOI: [10.15252/emmm.202012131](https://doi.org/10.15252/emmm.202012131)

Corresponding authors: Satish Madala (satish.madala@cchmc.org) , Anil Jegga (anil.jegga@cchmc.org)

Review Timeline:

Submission Date:	5th Feb 20
Editorial Decision:	24th Feb 20
Revision Received:	13th Jun 20
Editorial Decision:	3rd Jul 20
Revision Received:	8th Jul 20
Accepted:	13th Jul 20

Editor: Zeljko Durdevic

Transaction Report:

24th Feb 2020

Dear Dr. Madala,

Thank you for the submission of your manuscript to EMBO Molecular Medicine. We have now heard back from the three referees who agreed to evaluate your manuscript. As you will see from the reports below, all referees found the study timely/of interest/ important for the field [anything positive]. Still, while referee 3 is supportive of publication, referees 1 and 2 raise a number of concerns. We would be happy to consider a major revision of this work particularly addressing the following items:

- Rationale for investigating the role of Aurora Kinase B in idiopathic pulmonary fibrosis (IPF)
- Justification of the used animal model and confirmation of the key findings in the standard IPF mouse model.

Please note that addressing all the other points raised by the referees as much as possible will be necessary for further considering the manuscript in our journal, and acceptance of the manuscript will entail a second round of review. Considering the extent of the revision, I am happy to extend the revisions time to 6 months. EMBO Molecular Medicine encourages a single round of revision only and therefore, acceptance or rejection of the manuscript will depend on the completeness of your responses included in the next, final version of the manuscript. For this reason, and to save you from any frustrations in the end, I would strongly advise against returning an incomplete revision.

***** Reviewer's comments *****

Referee #1 (Remarks for Author):

In their manuscript, Kasam et al, explore a possible role for Aurora kinase B in fibroblast activation and the pathogenesis of idiopathic pulmonary fibrosis, both in vitro and in vivo. Moreover, pharmacological inhibition of Aurora B with basasertib attenuated fibrotic disease in an animal model, both in a prophylactic, as well as in a therapeutic scheme of administration. The topic is novel and up to date. Although this reviewer was convinced on a likely role of Aurora B in IPF, there are several theoretical and technical concerns mentioned below. Moreover, the included mechanistic insights, essentially modulation of proliferation and apoptosis, are way too general.

Major

Investigating a possible role of Aurora B in IPF was not well justified. According to the web site of the authors, "Barasertib, an AURKB-selective inhibitor, was identified using an integrative systems biology based approach and computational screening", and not vice versa. Maybe the paper should be completely restructured highlighting the potential of similar approaches.

The choice of the model, Tg-TGF α , over more widely used models should be explained.

The reported leakiness of the tetracycline model should be discussed; Doxycycline and 5% DMSO are both anti-inflammatory, that can affect lung pathogenesis even upon fibrotic stimuli. Therefore, control littermate groups (CCSP-rtTA alone +/- dox, +/- 5% DMSO, TetO-CMV-TGF α alone +/- Dox +/- 5% DMSO) should be used in all relative mouse studies, at least once. I am afraid that there is no easy way around the extreme number of mice necessary to fully appreciate the results with this model.

Although IPF can be referred as a fibroproliferative disease, the issue remains controversial and many researchers do not agree with such a definition. The consensus is that fibroblasts in IPF "persist" or "accumulate" rather than "proliferate higher" or "apoptose less". Please elaborate citing relevant high impact publications and/or recent reviews.

Is there an effect in mitotic functions of IPF fibroblasts that well fit with the known functions of Aurora B? Can the results be possibly explained by differences in the cell cycle? Is the circadian rhythm possible involved? Senescence?

Minor

Why all fibroblasts present with increased proliferation, while only a small fraction stained positive for Aurora B (Fig. 3C)? Please include a few references on increased proliferation, as well as decreased apoptosis, of lung fibroblasts upon TGF α (or TGF β , PDGF,...) stimulation.

Why the enzymatic inhibition of Aurora B leads to downregulation of its mRNA expression (Fig. 6C)?

The PK/PD profile of barasertib should be briefly mentioned to fully appreciate in vivo findings.

All graphs should be presented with scatter plots, rather than bar plots.

The number of repetitions for each experiment should be explicitly stated in each figure legend.

Negative controls (e.g isotype controls) should be included in a supplementary file for all IHC studies.

Additional supplementary figures should contain different magnifications of different parts of the fibrotic lung from mice and humans.

Fig. 1D. please show all 4 samples that were used for the quantification.

Sup Table 1 should include FCs, p, FDRs for all genes.

Referee #2 (Remarks for Author):

The study by Kasam et al demonstrates a novel pathological role for AurkB in fibroblast activation and pulmonary fibrosis. The major findings are that AurkB promotes myofibroblast activation and that pharmacological inhibition of this kinase effectively blocks development and progression of pulmonary fibrosis in mice. Investigators have also uncovered that Aurkb is regulated by WT1, which has important mechanistic implications. The manuscript is well written and findings are easily interpretable. The manuscript also incorporates data from human IPF tissues, increasing the clinical relevance of findings. Major criticisms are detailed below.

- 1) In figure 1, AurkB positive cells represent a small subset of mesenchymal cells in IPF tissues. This needs to be addressed somewhere in the manuscript. Perhaps a discussion of fibroblast heterogeneity could be added. Additionally it might be nice for investigators to quantify the number of AurkB positive and negative mesenchymal cells in tissues.
- 2) The TGF α is not a standard model of pulmonary fibrosis. The manuscript would be strengthened by using a standard model of pulmonary fibrosis, such as bleomycin.
- 3) Figure 7 does not show significant fibrosis in the untreated TGF α group. This needs to be addressed with better images.
- 4) Fibrosis in Figure 5 b subpleural TGF α image actually looks extrapleural. Can the authors confirm this is all subpleural?
- 5) Adventitial thickening is not a characteristic feature of pulmonary fibrosis. Please explain to the reader why this is relevant to IPF/ILD. This further illustrates the importance of using a standard model of pulmonary fibrosis.

Referee #3 (Remarks for Author):

I have been asked to review a very interesting manuscript by Drs Kasam and colleagues that reports on the role of Aurora Kinase B (AURKB) in fibroblast activation and pulmonary fibrosis. AURKB is associated with alignment and segregation of chromosomes during mitosis. The authors explore that AURKB might be important in fibroblast activation in pulmonary fibrosis. The author

show that stimulation of primary human lung fibroblasts by CTGF, TGF α , and IGF1 (but not TGF β) increases AURKB gene expression as measured at the mRNA level. In addition, higher levels of these transcripts were found in IPF fibroblasts compared to controls. It appears that more AURKB+ cells were observed by IH in IPF lungs. More AURKB (and not A) was observed in TGF α -overexpressing mice. The team then went on to knock down WT1 and show that silencing of WT1 decreased expression of AURKB. In addition overexpression of WT1 increased AURKB. Conserved WT1 sites were observed in the promoter region of both human and mouse AURKB. ChIP identified binding of a WT1 antibody to the AURKB gene, and a luciferase reporter was increased in WT1-overexpressing 293 cells. The team then employed a bioinformatic screen to look for overlapping genes in IPF lungs and siAURKB treated fibroblasts. Focusing on negative correlated genes, the team identified several potential AURKB-driven processes including cell proliferation, apoptosis, and ECM production.

This is a very solid manuscript which combines multiple modalities to establish the argument that AURKB is an important regulator of fibroblast activation in pulmonary fibrosis. Loss of- and gain of-function approaches complement a beautiful correlation with human gene expression data. This manuscript has potentially very high translational impact.

MAJOR COMMENTS

1. The majority of my comments relate to rationale. The team members are well known students of TGF α signaling and WT1 biology. Was AURKB identified by ChIP-Seq or RNA-Seq in a previous manuscript? I think the already strong impact of the manuscript would be enhanced by a strong rationale for pursuing AURKB.

MINOR COMMENTS

1. Many journals are requesting dot plots for data with fewer than N=10.

Point-by-point response to the reviewer comments:

Referee #1 (Remarks for Author):

In their manuscript, Kasam et al, explore a possible role for Aurora kinase B in fibroblast activation and the pathogenesis of idiopathic pulmonary fibrosis, both in vitro and in vivo. Moreover, pharmacological inhibition of Aurora B with barasertib attenuated fibrotic disease in an animal model, both in a prophylactic, as well as in a therapeutic scheme of administration. The topic is novel and up to date. Although this reviewer was convinced on a likely role of Aurora B in IPF, there are several theoretical and technical concerns mentioned below. Moreover, the included mechanistic insights, essentially modulation of proliferation and apoptosis, are way too general.

Response: We appreciate the reviewer comment regarding the novelty of our data on the role of Aurora kinase B (AURKB) in fibroblast activation and the use of AURKB inhibitor, barasertib, as a compound to modulate fibroproliferation, myofibroblast survival and ECM production. Also, we thank the reviewer for the additional comments that have helped us to improve the manuscript.

Major

Investigating a possible role of Aurora B in IPF was not well justified. According to the web site of the authors, "Barasertib, an AURKB-selective inhibitor, was identified using an integrative systems biology based approach and computational screening", and not vice versa. Maybe the paper should be completely restructured highlighting the potential of similar approaches.

Response: We now provide more insights on the rationale to investigate the role of AURKB in IPF and we believe that the rationale to support our hypothesis is two-fold. First, the current study on AURKB is a follow up study to our earlier publications on Wilms Tumor 1(WT1)-induced fibroblast activation in pulmonary fibrosis (PMID: 26371248 & 30135315). In particular, RNA-seq analysis of fibroblasts with the knockdown of WT1 identified AURKB as a potential downstream target of WT1 in mediating some, if not all, of the pro-fibrotic processes (PMID: 30135315). Second, we identified Tozasertib, a pan aurora kinase inhibitor as a potential anti-fibrotic candidate (PMID: 28239659). In this study, we queried IPF gene signatures using connectivity map approach against the LINCS database (<http://www.lincscloud.org/>), a massive catalog of gene-expression profiles collected from human cells treated with chemical and genetic perturbagens (PMID: 28239659). Based on findings from these two independent studies, we further characterized the expression of Aurora kinase isoforms in IPF and TGF α mouse model which we believe have helped us to develop a rationale to target AURKB using barasertib. We have now cited our previous findings and modified text accordingly in introduction of revised manuscript (Page 4; Lanes 13-19).

The choice of the model, Tg-TGF α , over more widely used models should be explained. The reported leakiness of the tetracycline model should be discussed; Doxycycline and 5% DMSO are both anti-inflammatory that can affect lung pathogenesis even upon fibrotic stimuli. Therefore, control littermate groups (CCSP-rtTA alone +/- dox, +/- 5% DMSO,

TetO-CMV-TGF α alone +/- Dox +/- 5% DMSO) should be used in all relative mouse studies, at least once. I am afraid that there is no easy way around the extreme number of mice necessary to fully appreciate the results with this model.

Response: Published studies have shown increased levels of TGF α and other EGFR ligand in the lung lavage fluid of IPF patients compared to healthy controls (PMID: 10207942). EGFR has been shown activated by several profibrotic agents including TGF β , TNF α , and IL-13 (PMID:23086930; PMID:11726400; PMID:10623834). In support, overexpression of TGF α in mice has resulted in the development of pathologic fibrotic lesions similar to those found in IPF, including increased fibroproliferation, survival and subpleural fibrosis migrating into the interstitium, differentiation of myofibroblasts, and formation of fibroblastic foci (PMID:20676040; PMID:31156440). Moreover, gene expression profiles after expression of TGF α were more similar to IPF samples compared to that of a bleomycin model (PMID: 17496152; PMID: 31156440). Therefore, the TGF α -TG mouse is a reliable preclinical model to assess fibroblast activation involved in pulmonary fibrosis. Also, we were able to replicate our findings with a TGF α model on barasertib therapy using a widely used mouse model of bleomycin-induced pulmonary fibrosis (Figure 8). We included changes in the text with the details on relevance of TGF α model to study pulmonary fibrosis (Page 17; Lanes 3-19).

We agree with the reviewer that off-target effects of rtTA and Dox are important factors to consider the effects of transgene and anti-fibrotic therapy (PMID:20676040; PMID:22180870). As suggested by the reviewer, we now performed in vivo studies to assess possible anti-inflammatory effects of Doxycycline and DMSO using appropriate control mice (CCSP-rtTA mice and TetO-CMV-TGF α mice). In agreement with our previous studies, no gross changes in inflammation or fibrosis were observed with the long-term treatment of DMSO or doxycycline (Appendix Fig S2).

Although IPF can be referred as a fibroproliferative disease, the issue remains controversial and many researchers do not agree with such a definition. The consensus is that fibroblasts in IPF "persist" or "accumulate" rather than "proliferate higher" or "apoptosis less". Please elaborate citing relevant high impact publications and/or recent reviews. Is there an effect in mitotic functions of IPF fibroblasts that well fit with the known functions of Aurora B? Can the results be possibly explained by differences in the cell cycle? Is the circadian rhythm possible involved? Senescence?

Response: We agree with the reviewer that the mature fibrotic lung lesions show limited proliferation as these lesions are dominated by apoptosis resistant myofibroblasts that produce increased amounts of collagen. However, it has been observed that there is a significant increase in the number of proliferating fibroblasts predominantly in the early fibrotic lung lesions including expanding areas of fibrotic foci (PMID: 28530639). Although fibroproliferation may play a more diminished role in established fibrosis, it may still play an important role in early fibrotic lesions, where expansion and accumulation of fibroblasts is occurring. IPF lung tissue is histologically heterogenous containing normal-appearing parenchyma with early fibrotic lesions of thickened alveolar walls adjacent to advanced scar tissue with honeycombing and bronchiolization (PMID: 25217476). Recent single cell RNA seq studies further support molecular and functional heterogeneity in stromal cells

that accumulate in these fibrotic lesions (PMID: 32317643; PMID: 30554520; PMID: 29590628). Our new findings suggesting that barasertib therapy is effective in reducing fibrosis burden in part could be due to its ability to block both fibroproliferation and myofibroblast survival. Therefore, AURKB inhibition by barasertib is more effective in reducing early and established fibrotic lung lesions.

Mitosis is a highly regulated process in which AURKB plays an essential role by orchestrating connections between spindle microtubules and kinetochores, facilitating the proliferation. With the progression of fibrosis, the local fibroblasts transform to myofibroblasts that had initially proliferated, in part mediated by AURKB. Myofibroblasts in the mature fibrotic lesions have been shown to acquire a senescent phenotype that might play a pathogenic role in pulmonary fibrosis (PMID:28230051). We have not assessed whether AURKB-positive myofibroblasts that accumulate in fibrotic lesions exhibit any molecular features of cellular senescence. However, our RNA seq analysis of AURKB-deficient fibroblasts revealed significant transcriptomic differences, particularly in genes associated with proliferation, apoptosis, and ECM, suggesting that AURKB could be a fibrogenic factor with a limited role in myofibroblast senescence. In the revision, we have expanded our discussion on possible AURKB-driven effects on cell cycle, apoptosis and senescence (Page 18; Lanes 10-17).

Minor

Why all fibroblasts present with increased proliferation, while only a small fraction stained positive for Aurora B (Fig. 3C)? Please include a few references on increased proliferation, as well as decreased apoptosis, of lung fibroblasts upon TGF α (or TGF β , PDGF,...) stimulation.

Response: Fewer AURKB-positive fibroblasts could be due to differences in the expression of Ki67 and AURKB in dividing cells. While Ki67 is expressed during all active phases of the cell cycle (G1, S, G2 and mitosis), AURKB expression increases selectively during G2-M transition. Also, other factors might be mediating fibroproliferation in a subset of fibroblasts supporting their molecular heterogeneity. We now include references that support the role of TGF α and other growth factors in increased proliferation and reduced apoptosis of fibroblasts (PMID: 11524244).

Why the enzymatic inhibition of Aurora B leads to downregulation of its mRNA expression (Fig. 6C)?

Response: We assessed WT1 levels that involved in AURKB expression in fibroblasts treated with AURKB inhibitor. Our new data suggest a significant decrease in WT1 transcripts with AURKB inhibition (Fig EV5). However, we do not know mechanisms underlying the observed decrease in WT1 transcripts with AURKB inhibition. Future studies are therefore warranted to identify mechanisms in barasertib-regulated AURKB/WT1 expression.

The PK/PD profile of barasertib should be briefly mentioned to fully appreciate in vivo findings.

Response: We have revised the manuscript to describe the PK/PD profile of barasertib (Page 20; Lanes 1-6).

All graphs should be presented with scatter plots, rather than bar plots.

Response: We have changed all bar graphs to scatter plots in the revised manuscript.

The number of repetitions for each experiment should be explicitly stated in each figure legend.

Response: Appropriate changes were made in the revised manuscript.

Negative controls (e.g isotype controls) should be included in a supplementary file for all IHC studies.

Response: As suggested images with isotype control staining are included (Appendix Fig S4)

Additional supplementary figures should contain different magnifications of different parts of the fibrotic lung from mice and humans.

Response: As suggested by the reviewer, we now include low magnification images in Appendix Fig S3.

Fig. 1D. Please show all 4 samples that were used for the quantification.

Response: We now provide data on additional samples used for the quantification of Aurkb and Aurka as Appendix Fig S1.

Sup Table 1 should include FCs, p, FDRs for all genes.

Response: We now include suggested changes to the table S1 in the revised manuscript.

Referee #2 (Remarks for Author):

The study by Kasam et al demonstrates a novel pathological role for AurkB in fibroblast activation and pulmonary fibrosis. The major findings are that AurkB promotes myofibroblast activation and that pharmacological inhibition of this kinase effectively blocks development and progression of pulmonary fibrosis in mice. Investigators have also uncovered that Aurkb is regulated by WT1, which has important mechanistic implications. The manuscript is well written and findings are easily interpretable. The manuscript also incorporates data from human IPF tissues, increasing the clinical relevance of findings. Major criticisms are detailed below.

Overall Response: We thank the reviewer for suggestions that have substantially helped us to improve the revised version of our manuscript.

1) In figure 1, AurkB positive cells represent a small subset of mesenchymal cells in IPF tissues. This needs to be addressed somewhere in the manuscript. Perhaps a discussion of fibroblast heterogeneity could be added. Additionally it might be nice for investigators to quantify the number of AurkB positive and negative mesenchymal cells in tissues.

Response: We agree with the reviewer that a subset of mesenchymal cells express Aurora B (Fig 1C) and we now discussed the heterogeneity of fibroblasts including AURKB-positive fibroblasts in the revised manuscript (Page 18; Lanes 13-17). Unfortunately, we did not quantify the number of AURKB-positive mesenchymal cells due to the lack of co-staining with mesenchymal cell markers.

2) The TGF α is not a standard model of pulmonary fibrosis. The manuscript would be strengthened by using a standard model of pulmonary fibrosis, such as bleomycin.

Response: We would like to thank the reviewer for the suggestion to measure the effects of barasertib using bleomycin model. We have carried out several additional experiments to further test in vivo efficacy of barasertib to attenuate ongoing fibrosis in a chronic mouse model of bleomycin-induced pulmonary fibrosis. These studies showed that barasertib attenuates collagen deposition (Fig 8). Also, we observed a significant decrease in the expression of genes involved in fibroblast activation. When viewed in combination with in vivo data using TGF α model, these data are consistent with the concept that AURKB inhibition with barasertib is an effective therapy to attenuate fibroblast activation and pulmonary fibrosis.

3) Figure 7 does not show significant fibrosis in the untreated TGF α group. This needs to be addressed with better images.

Response: We now include modified images to reflect fibrosis in the untreated TGF α group (Fig 7D).

4) Fibrosis in Figure 5 b subpleural TGF α image actually looks extrapleural. Can the authors confirm this is all subpleural?

Response: Our published studies have demonstrated that these lesions are subpleural and express mesenchymal markers but are negative for mesothelial cell markers (PMID: 30135315).

5) Adventitial thickening is not a characteristic feature of pulmonary fibrosis. Please explain to the reader why this is relevant to IPF/ILD. This further illustrates the importance of using a standard model of pulmonary fibrosis.

Response: We agree with the reviewer and understand the limitations with preclinical models available to study IPF. We performed additional studies on testing barasertib using a chronic mouse model of bleomycin-induced pulmonary fibrosis (PMID:28775096). These new data suggest a significant decrease bleomycin-induced pulmonary fibrosis in

mice treated with barasertib compared to vehicle treated control mice. These new findings have been clearly stated in the revised version (Fig 8).

Referee #3 (Remarks for Author):

I have been asked to review a very interesting manuscript by Drs Kasam and colleagues that reports on the role of Aurora Kinase B (AURKB) in fibroblast activation and pulmonary fibrosis. AURKB is associated with alignment and segregation of chromosomes during mitosis. The authors explore that AURKB might be important in fibroblast activation in pulmonary fibrosis. The authors show that stimulation of primary human lung fibroblasts by CTGF, TGF α , and IGF1 (but not TGF β) increases AURKB gene expression as measured at the mRNA level. In addition, higher levels of these transcripts were found in IPF fibroblasts compared to controls. It appears that more AURKB+ cells were observed by IH in IPF lungs. More AURKB (and not A) was observed in TGF α -overexpressing mice. The team then went on to knock down WT1 and show that silencing of WT1 decreased expression of AURKB. In addition overexpression of WT1 increased AURKB. Conserved WT1 sites were observed in the promoter region of both human and mouse AURKB. ChIP identified binding of a WT1 antibody to the AURKB gene, and a luciferase reporter was increased in WT1-overexpressing 293 cells. The team then employed a bioinformatic screen to look for overlapping genes in IPF lungs and siAURKB treated fibroblasts. Focusing on negative correlated genes, the team identified several potential AURKB-driven processes including cell proliferation, apoptosis, and ECM production. This is a very solid manuscript which combines multiple modalities to establish the argument that AURKB is an important regulator of fibroblast activation in pulmonary fibrosis. Loss of- and gain of-function approaches complement a beautiful correlation with human gene expression data. This manuscript has potentially very high translational impact.

Response: We thank the reviewer for the comprehensive review and positive feedback on our manuscript.

MAJOR COMMENTS

1. The majority of my comments relate to rationale. The team members are well known students of TGF α signaling and WT1 biology. Was AURKB identified by ChIP-Seq or RNA-Seq in a previous manuscript? I think the already strong impact of the manuscript would be enhanced by a strong rationale for pursuing AURKB.

Response: This is an excellent suggestion and we revised our manuscript to highlight the rationale behind this study. In particular, our RNA-seq analysis of transcripts with the knockdown of WT1 have identified AURKB as a potential WT1 target gene involved in fibroblast activation. Also, our recent study using pharmacogenomic screens identified AURK inhibitors to attenuate fibroblast activation in IPF. As further discussed in the response to Reviewer 1 (Major point#1), we have significantly revised our text in the introduction to highlight the rationale for pursuing AURKB (Pages 4-5).

MINOR COMMENTS

1. Many journals are requesting dot plots for data with fewer than $N=10$.
Response: Data are changed to dot plots in the revised manuscript as suggested by the reviewer.

3rd Jul 2020

Dear Dr. Madala,

Thank you for the submission of your revised manuscript to EMBO Molecular Medicine. I am pleased to inform you that we will be able to accept your manuscript pending the following final amendments.

***** Reviewer's comments *****

Referee #1 (Remarks for Author):

no further comments

Referee #2 (Remarks for Author):

The authors have addressed all previous criticisms.

We like to thank editor for accepting our manuscript for publication in EMBO molecular medicine. As suggested, we now include modifications needed to the text and also high resolution images.

The authors performed the requested changes.

Corresponding Author Name: Satish K Madala

Manuscript Number: EMM-2020-12131